# Overall Complication Rates of DIEP Flap Breast Reconstructions in Germany—A Multi-Center Analysis Based on the DGPRÄC Prospective National Online Registry for Microsurgical Breast Reconstructions

**DOI:** 10.3390/jcm10051016

**Published:** 2021-03-02

**Authors:** Paul I. Heidekrueger, Nicholas Moellhoff, Raymund E. Horch, Jörn A. Lohmeyer, Mario Marx, Christoph Heitmann, Hisham Fansa, Matthias Geenen, Christian J. Gabka, Steffen Handstein, Lukas Prantl, Uwe von Fritschen

**Affiliations:** 1Centre of Plastic, Aesthetic, Hand and Reconstructive Surgery, University of Regensburg, 93053 Regensburg, Germany; 2Division of Hand, Plastic and Aesthetic Surgery, University Hospital, LMU Munich, 80336 Munich, Germany; nicholas.moellhoff@med.uni-muenchen.de; 3Department of Plastic and Hand Surgery, University Hospital of Erlangen, 91054 Erlangen, Germany; raymund.horch@uk-erlangen.de; 4Department of Plastic, Reconstructive and Aesthetic Surgery, Agaplesion Diakonieklinikum Hamburg, 20259 Hamburg, Germany; joern.lohmeyer@d-k-h.de; 5Department of Plastic, Reconstructive and Breast Surgery, Elbland Hospital Radebeul, 01445 Radebeul, Germany; mario.marx@elblandkliniken.de; 6SENO Clinic, 80539 Munich, Germany; info@seno-mvz.de; 7Department of Plastic, Reconstructive, and Aesthetic Surgery, Breast Centre Spital Zollikerberg, 8125 Zollikerberg, Switzerland; plast.chir@spitalzollikerberg.ch; 8Department of Reconstructive Surgery, Lubinus Clinic Kiel, 24106 Kiel, Germany; m.geenen@lubinus-clinicum.de; 9Nymphenburg Clinic for Plastic and Aesthetic Surgery, 80636 Munich, Germany; mail@gabka-spiegel.de; 10Department of Plastic, Reconstructive, and Breast Surgery, Municipal Hospital Goerlitz, 02828 Görlitz, Germany; plastische.chirurgie@klinikum-goerlitz.de; 11Department of Plastic and Aesthetic Surgery, Hand Surgery, Helios Hospital Emil von Behring, 14165 Berlin, Germany

**Keywords:** breast reconstruction, deep inferior epigastric perforator (DIEP) flap, breast cancer, microsurgery, plastic surgery, reconstructive surgery

## Abstract

While autologous breast reconstruction has gained momentum over recent years, there is limited data on the structure and quality of care of microsurgical breast reconstruction in Germany. Using the breast reconstruction database established by the German Society of Plastic, Reconstructive and Aesthetic Surgeons (DGPRÄC), the presented study investigated the overall outcomes of deep inferior epigastric perforator (DIEP) flap reconstructions in Germany. Data of 3926 patients and 4577 DIEP flaps performed by 22 centers were included in this study. Demographics, patient characteristics, perioperative details and postoperative outcomes were accounted for. Centers performing < Ø 40 (low-volume (LV)) vs. ≥ Ø 40 (high-volume (HV)) annual DIEP flaps were analyzed separately. Overall, total and partial flap loss rates were as low as 2.0% and 1.1% respectively, and emergent vascular revision surgery was performed in 4.3% of cases. Revision surgery due to wound complications was conducted in 8.3% of all cases. Mean operative time and length of hospital stay was significantly shorter in the HV group (LV: 385.82 min vs. HV: 287.14 min; LV: 9.04 (18.87) days vs. HV: 8.21 (5.04) days; both *p* < 0.05). The outcome and complication rates deduced from the national registry underline the high standard of microsurgical breast reconstruction on a national level in Germany.

## 1. Introduction

Since its first description in 1979 [1], autologous breast reconstruction has evolved as a safe and viable option. More than that, it is now regarded as the international gold-standard in reconstructive breast surgery [2,3]. Breast reconstruction was revolutionized in 1989, when Koshima et al. introduced perforator-based reconstruction using the deep inferior epigastric perforator (DIEP) flap [4], thus significantly reducing donor site morbidity whilst at the same time maximizing clinical outcomes and generating aesthetically pleasing results. Since then, the DIEP flap has emerged as a workhorse in reconstructive breast surgery [5,6,7,8,9,10].

Currently, breast reconstruction is performed either by plastic surgeons or gynecologists. Both specialties often differ significantly in regard to the reconstructive approach preferred, and no clear international or interdisciplinary guidelines exist to support the decision-making process. While superior results are reported for autologous breast reconstruction in terms of aesthetic and natural outcome, longevity of postoperative results, long-term patient satisfaction and quality of life [11,12,13], implant-based reconstructions are still, by far, the most commonly performed procedure after breast cancer surgery. In fact, rates of autologous reconstruction have continued at a stable, rather than increasing level, for many years [14,15,16].

For various reasons, the introduction of microsurgical breast reconstruction into patient care has yet to gain strong momentum in Germany [17]. To further improve the standpoint of modern microsurgical reconstructive procedures within interdisciplinary treatment concepts, the German Society of Plastic, Reconstructive and Aesthetic Surgeons (DGPRÄC) introduced a national online registry to underline the structure and quality of care of microsurgical breast reconstruction in Germany.

Using this prospectively maintained free flap breast reconstruction database, the presented study investigated overall complication rates of DIEP flap breast reconstructions in Germany, based on a large patient cohort of 3926 female patients. By demonstrating flap success rates of over ~97%, this study underlines that successful microsurgical breast reconstruction can be achieved on a national level as in Germany, as described in international literature.

## 2. Materials and Methods

Patient data were obtained from the DGPRÄC national online registry for microsurgical breast reconstructions, including data sets from January 2011–July 2018. Parts of this database have been previously investigated by our study group [18,19,20,21,22]. Fritschen et al. have described the purpose and design of the registry in detail [17]. All elements of the study were performed in accordance with institutional guidelines and regulations. Ethical board approval was obtained prior to study initiation from the Bavarian State Medical Association (156/17 S) and the Berlin Chamber of Physicians (Eth-V-Q/17)). Patient data were entered anonymously.

Data were entered prospectively, sorted and tagged to the individual plastic-surgical department and surgeon. Data acquired included clinical outcome, relevant individual patient parameters and characteristics, as well as prior therapeutic steps such as systemic breast cancer therapy or previous surgery. Intra- and perioperative details as well as data regarding surgical technique were also entered. Outcome was assessed and follow-up data generated up to three months postoperatively.

The database was open to plastic surgical units in Germany. To assure high quality conclusions, only data points from centers previously certified by the DGPRÄC according to specifically defined criteria (Table 1) were analyzed in the study.

Audit monitoring visits were performed by qualified monitors according to a defined protocol and checklist, in order to ascertain the quality of the entered data in comparison to the hospital’s documentation.

### 2.1. Patient Cohort

To investigate a homogenous group of breast reconstructions, only female patients receiving uni- or bilateral DIEP flap breast reconstruction were included in this study. Thus, data of 3926 patients and 4577 DIEP flaps were included in this study. The data were generated by a total of 22 plastic surgical centers. Cases were divided into two groups, depending on the experience of the centers with DIEP flap surgery: a high-volume group (≥Ø 40 DIEP flaps performed per year of data entry) vs. a low-volume group (<Ø 40 DIEP flaps performed per year of data entry). Surgical complications were then compared between both groups. The following outcomes were investigated: total flap loss, partial flap loss (10–20% of free flap tissue), unexpected or emergency revision surgery (vascular revision, i.e., arterial and venous thrombosis; wound revision, i.e., infection, hematoma and wound healing disturbances at donor and recipient site) and medical complications.

### 2.2. Statistical Analysis

Differences between groups were determined using ANOVA or a chi-squared test of independence. Statistical analyses were performed using SAS (Version 9.4, The SAS institute, Cary, NC, USA), and results were considered statistically significant for values of *p* ≤ 0.05 to guide conclusions.

## 3. Results

### 3.1. Descriptive Data

Descriptive and demographic data is summarized in Table 2. The investigated study sample consisted of 3926 female patients, with a mean age of 51.30 (SD 31.61) years and a mean body mass index (BMI) of 26.28 (SD 4.44) kg/m^2^, who received 4577 free DIEP flap breast reconstructions. Within this total, 3236 free flaps (70.7%) were unilateral breast reconstructions, while 1341 flaps (29.3%) were bilateral breast reconstructions. Immediate breast reconstructions were performed in 24.8% of cases.

Reconstructions were performed at 22 plastic-surgical centers. Five of these centers performed an average of ≥40 DIEP flaps per year of data entry and were thus classified as high-volume clinics. A majority of 17 centers performed <40 DIEP flaps per year and were thus classified as low-volume clinics.

The number of patients who reported a history of smoking was 476 (10.4%). Further details with regard to number of pack years were not available. Patient comorbidities assessed within the database showed a total of 125 (2.7%) cases with diagnosed diabetes mellitus. Cases showing a clinical history of deranged hemostasis with impaired clot formation amounted to 71 (1.6%), and 192 (4.2%) free flaps were performed in patients with a prior abdominal scar >10 cm. A positive family history of breast and/or ovarian cancer in first degree relatives (FDRs) was found in 1191 (26.0%) cases, and 697 (15.2%) cases were associated with a genetic disposition for breast cancer.

More than half of all cases (2605, 56.9%) received chemotherapy, whereas 2206 (48.2%) of all cases received chemotherapy within six months prior to breast reconstruction. Immunosuppressive therapy using targeted antibodies was only administered in 34 (0.7%) cases, whereas 484 (10.6%) cases received Tamoxifen therapy prior to surgery.

The most common indication for breast reconstruction was status after mastectomy (40.5%), followed by DIEP flap breast reconstruction after complications associated with other reconstructive techniques (21.2%; including implant or other flap type-based reconstructions), primary carcinoma (11.4%) and status after breast conserving therapy (8.4%). Prophylactic mastectomy due to positive familial history certified by a genetic test accounted for 6.8% of all reconstructions.

### 3.2. Perioperative Details

The mean duration of DIEP flap reconstruction was 318.60 (SD 127.94) minutes and the mean ischemia time was 50.81 (SD 25.86) minutes. The internal mammary artery was used for anastomosis in the majority of cases (3683 flaps, 80.5%). Perioperative antibiotics (single-shot) were administered in almost all cases (4399, 96.1%). Mobilization was begun at the first postoperative day in almost three quarters of all cases (3293, 72.0%). Mean length of hospital stay (LOS) was 8.47 (SD 11.42) days (Table 3).

### 3.3. Postoperative Complications and Comparison of Low- and High-Volume Centers

The overall flap success rate was 96.9%. Total flap loss was seen in 92 (2.0%) cases, whereas partial flap loss occurred in 51 (1.1%) cases. Emergent vascular revision surgery was performed in 4.3% of cases. In 2.7% of all cases, vascular revision surgery was necessary due to venous thrombosis, compared to 1.6% of cases with arterial thrombosis. Revision surgery due to wound complications was necessary in 8.3% of all cases, with hematoma at the recipient site being the most common reason (3.2%), followed by wound-healing disturbances at the donor (1.7%) and recipient site (1.5%). Medical complications occurred in 294 (6.4%) cases (Table 4).

To evaluate potential differences in complication rates with regard to the volume of DIEP flaps performed and thus the experience of the individual centers with this operative technique, we separately evaluated low vs. high-volume centers in detail. The low-volume (LV) group included 1260 female patients receiving 1459 DIEP flaps, and the high-volume (HV) group included 2653 female patients receiving 3118 DIEP flaps (Table 5). Mean operation time was significantly shorter in the HV group (LV: 385.82 min vs. HV: 287.14 min; *p* < 0.001). Mean ischemia time, however, was comparable between both groups (*p* = 0.073). Postoperative mobilization commenced significantly earlier in the HV group (*p* < 0.001). In addition, LOS was significantly shorter in the HV group (LV: 9.04 (18.87) days vs. HV: 8.21 (5.04) days; *p* = 0.023) (Table 5).

Interestingly, outcome analysis showed a significantly higher rate of total flap loss within the HV-group (LV: 1.2% vs. HV: 2.4%; *p* = 0.014), whereas partial flap loss was comparable between groups (*p* = 0.327). Emergent vascular revision was performed in 4% (LV) and 4.5% (HV) of cases, without showing statistical significance (*p* = 0.453). Revision surgery due to wound complications was, however, performed significantly more often in the LV group (LV: 10.8% vs. HV: 7.1%; *p* < 0.001). This was related to a significantly higher number of wound-healing disturbances at the recipient site requiring revision surgery in this group (LV: 2.7% vs. HV: 1.0%) (Table 6).

## 4. Discussion

Despite the evident advantages of autologous breast reconstruction, it is often underrepresented as treatment of choice for many breast cancer patients. Several reasons might account for this phenomenon: first, autologous breast reconstruction is associated with higher procedural and hospital costs, due to operative time, longer postoperative morbidity and hospital stay, although recent studies found similar long-term total cost of care compared to implant-based reconstruction [23,24]. Second, patient awareness and information of the different reconstructive options available plays an important role and may be in need of improvement [25]. In Germany, the therapy of breast cancer is often integrated into specialized interdisciplinary breast cancer centers, which are most commonly managed by gynecologists. While the certification criteria stipulate that patients must be offered all available reconstructive techniques, by providing plastic-surgical care if needed, implementation into clinical practice still needs significant improvement [17,26]. In line with this, Albornoz et al. identified that sociodemographic variables and hospital characteristics may influence the method of breast reconstruction [27]. Last but not least, gynecologically managed breast cancer centers have doubted the quality of care of microsurgical breast reconstruction in Germany and questioned whether it can meet international standards and outcomes particularly outside of specialized centers in a national setting. While international studies show increasingly good outcome rates after autologous breast reconstructions, reaching excellent levels of an average flap loss rates of up to 3%, [28] no data existed concerning the situation of care in Germany.

The DGPRÄC online registry was designed to tackle this problem and to shed further light onto the structure and the quality of microsurgical autologous breast reconstruction in Germany. Overall, the presented data show that free DIEP transfer for breast reconstruction is performed under high quality standards over a broad number of centers in Germany, with total and partial flap loss rates as low as 2.0% and 1.1% respectively, and emergent vascular revision surgery being performed in 4.3% of cases. The outcome and complication rates deduced from the national registry compare to recent large-scale international studies and show no significant disparity in this regard [29,30,31,32]. The study population of Vemula et al. (478 DIEP flaps) showed overall DIEP flap success rates of 98.2% at specialty surgery hospitals and 96.4% at tertiary care facilities [29]. Unukovych et al. found a reoperation rate of 15.9% in a study population of 503 DIEP flaps. Mirroring our results, flap failure was encountered in 2.0% of cases, while partial flap loss was found in 1.2% of all cases [30]. Depypere et al. report revision surgery in 5% (48/965) of all DIEP flaps investigated [31]. The study group of Vanschoonbeek et al. investigated a total of 1330 DIEP free flaps, of which 3.38% required urgent exploration. In accordance with our study results, venous insufficiency was the main reason for the revision of DIEP flaps. The flap failure rate reported was 1.28%, thus being slightly lower than the one presented in this manuscript [32]. Notably, these studies investigated a study population significantly smaller than the one presented within this manuscript.

The descriptive data and demographical variables of the investigated study population were comparable to international literature. For example, in our study the mean age of patients receiving DIEP flap transfer was 51.3 years, compared to a median age of 51.6 years in the study population investigated by Kamali et al. [16] and 46.7 years in the study population evaluated by Depypere et al. [31]. Mean BMI was 26.28 kg/m^2^ and therefore slightly higher in comparison to the population investigated Vanschoonbeek et al. (24.9 kg/m^2^) [32] and almost equal to the mean BMI of the study population investigated by Unukovych et al. (26.2 kg/m^2^) [30]. Neoadjuvant chemotherapy was administered in approximately 50% of all cases, complying with numbers found in the studies of Vemula et al. [29] and Unukovych et al. [30]. Overall, immediate breast reconstruction (IBR) was performed in ~25% of all cases which seems relatively low compared to the international literature [14,33,34]. Several studies have reported a continuous increase in IBR, which, however, can also be traced back to an increased amount of implant-based immediate breast reconstructions [35]. Importantly, our data show a significant increase in autologous IBR between the years 2011 and 2018 (Figure 1; *p* < 0.001 between years and type of reconstruction, immediate vs. delayed breast reconstruction).

Compared to international standards, especially in the U.S., LOS was significantly longer in the patient population investigated in this study. While patients after DIEP flap reconstruction stay hospitalized for 3–4 days in the U.S. [35], we found the LOS in our study to be as high as 8.47 (SD 11.42) days. This complies with findings of Ridic et al. who compared overall LOS in health care systems in the U.S. and Germany. According to their study, the average LOS in Germany is generally much longer than in the United States (12.0 vs. 7.1 days) which can be related to the significantly larger capacity in the number of hospital beds relative to the population [36]. In addition, the structure of outpatient care and overall remuneration differs largely between the respective healthcare systems, with the DRG-specific medium length of stay being 9.1 days for breast cancer patients receiving microsurgical autologous reconstruction in Germany.

Importantly, when comparing high- vs. low-volume centers, the data presented in this manuscript show a significant reduction of LOS for centers performing more than an average of 40 DIEP flaps annually. Probably, this relates to the significantly earlier postoperative mobilization and shorter operation time by the factor of 0.74 observed in these centers. Unexpectedly, ischemia time was comparable between the HV and LV group, rather than being significantly shorter within the HV group. The data suggest that the reduced operation time could result from significantly quicker and more efficient flap harvest rather than faster microsurgical anastomosis at the recipient site. Additionally, centers performing a lower volume of flaps might have slimmer personnel structures, prohibiting a two-team approach during flap harvest and recipient vessel preparation, thus accounting for prolonged operative times. Surprisingly, while generally being at a low ~2%, flap loss rates were slightly higher in HV-centers. The data provides no conclusive evidence to explain this result, which is why future studies are needed to elaborate on this finding. Albornoz et al. found an inverse relationship between centers immediate autologous breast reconstruction volume and overall complications [37], specifically with regards to surgery-related complications. Unfortunately, the study group performed no detailed investigation comparing total and partial flap loss, as well as revision rates between groups, which limits comparability to the data presented within this manuscript.

The short follow-up time of three months postoperatively must be considered a limitation of this study. Significant complications, such as donor-site bulge and hernia could thus not be accounted for sufficiently. Studies with an extended follow-up of 12 months or more are needed in order to further define the long-term complications associated with free DIEP flap breast reconstruction.

By publicizing and evaluating data from the DGPRÄC national online registry for microsurgical breast reconstruction, the authors hope to increase visibility and transparency of the standard and quality of care in breast reconstruction. By doing so, and in the best interests of our patients, we aim to improve the cooperation between oncologic and reconstructive surgeons, increase patients’ awareness of the reconstructive measures available and strengthen the role of microsurgical breast reconstruction in breast cancer treatment.

## 5. Conclusions

This study investigated a large national database of DIEP flap breast reconstructions initiated by the German Society of Plastic, Reconstructive and Aesthetic Surgeons (DGPRÄC). Breast reconstruction using microvascular free DIEP flap transfer is performed under high quality standards over a broad number of plastic-surgical centers in Germany. Complications, as well as the rates of total and partial flap loss, are low, and outcomes compare to large-scale international studies, thus underlining the structure and quality of care of microsurgical breast reconstruction in Germany.

## Figures and Tables

**Figure 1 jcm-10-01016-f001:**
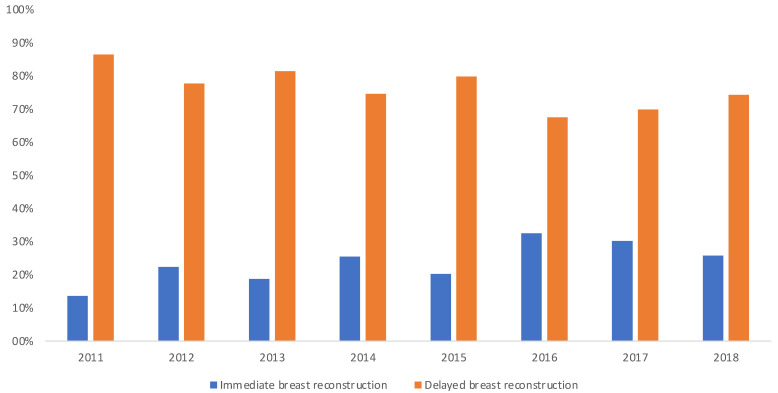
Increasing trend of immediate breast reconstruction performed between 2011–2018, pooled across all 22 centers (*p* < 0.001 between years and type of reconstruction, immediate vs. delayed breast reconstruction).

**Table 1 jcm-10-01016-t001:** Criteria defined by the German Society of Plastic, Reconstructive and Aesthetic Surgeons (DGPRÄC) for the certification of plastic-surgical centers that entered data into the national online registry for microsurgical breast reconstructions.

Criteria for DGPRÄC Certification
Centers must perform at least 100 annual breast procedures
At least 20 microsurgical breast reconstructions must be performed by a single surgeon
Five of these procedures may be performed as teaching operations
Procedures performed by an assistant surgeon (aside from teaching operations) are not counted

Criteria translated from German, originally published in Fritschen et al. [17].

**Table 2 jcm-10-01016-t002:** Demographics and patient characteristics.

Variable	
Patients, *n*	3926
Free flaps, *n*	4577
Age, years	
Mean (SD)	51.30 (31.61)
BMI, kg/m^2^	
Mean (SD)	26.28 (4.44)
Immediate reconstruction, *n* (%)	1136 (24.8)
Secondary reconstruction, *n* (%)	3441 (75.2)
Reconstructed side, *n* (%)	
Right	1560 (34.1)
Left	1676 (36.6)
Both	1341 (29.3)
Smoking history, *n* (%)	476 (10.4)
Comorbidities, *n* (%)	
Diabetes mellitus	125 (2.7)
Coagulopathy *	71 (1.6)
Abdominal scar >10 cm, *n* (%)	192 (4.2)
Family history of breast and/or ovarian cancer in FDRs, *n* (%)	1191 (26.0)
Genetic disposition, *n* (%) **	697 (15.2)
Chemotherapy within the last 6 months, *n* (%) ***	2605 (56.9)
Chemotherapy prior to the last 6 months, *n* (%) ****	2206 (48.2)
Immunosuppressive therapy, *n* (%) +	34 (0.7)
Tamoxifen therapy, *n* (%) ++	484 (10.6)
Indication, *n* (%)	
Status after mastectomy	1555 (40.5)
DCIS	180 (4.7)
Primary carcinoma	436 (11.4)
Familial risk +++	262 (6.8)
Complications after prior reconstructive procedures ++++	813 (21.2)
Benign tumor	47 (1.2)
Status after BCT	321 (8.4)
Tumor recurrence	122 (3.2)
other	105 (2.7)

Percentages calculated based on the number of free flaps. * Clinical history of deranged hemostasis; ** family history of breast cancer without a positive genetic test; *** chemotherapy less than 6 months prior to DIEP flap; **** chemotherapy more than 6 months prior to DIEP flap; + immunotherapy with targeted antibodies; ++ Tamoxifen therapy for women with hormone-receptor positive breast cancer; +++ risk-reducing mastectomy performed in patients with a genetic mutation for familial/hereditary breast cancer; ++++ complications after previous reconstructive breast cancer surgery (i.e., implant, other pedicled/free flap transfer). *n*, number; SD, standard deviation; BMI, body mass index; FDR, first degree relatives; BCT, breast conserving therapy; DCIS, ductal carcinoma in situ.

**Table 3 jcm-10-01016-t003:** Perioperative details.

Variable	
Free flaps, *n*	4577
Operation time, min	
Mean (SD)	318.60 (127.94)
Ischemia time, min	
Mean (SD)	50.81 (25.86)
Recipient, *n* (%)	
Internal mammary	3683 (80.5)
Thoracodorsal	704 (15.4)
Other	190 (4.2)
Perioperative antibiotics, *n* (%)	4399 (96.1)
Postoperative mobilization, *n* (%)	
Day 1	3293 (72.0)
Day 2	773 (16.9)
Day 3	126 (2.8)
Day >3	378 (8.3)
LOS, days	
Mean (SD)	8.47 (11.42)

Percentages calculated based on the number of free flaps. *n*, number; SD, standard deviation; min, minutes; LOS, length of hospital stay.

**Table 4 jcm-10-01016-t004:** Postoperative complications and free flap outcome over a follow-up period of three months. Percentages calculated based on the number of free flaps.

Variable	
Free flaps, *n*	4577
Total flap loss, *n* (%)	92 (2.0)
Partial flap loss, *n* (%)	51 (1.1)
Emergent vascular revision surgery, *n* (%)	197 (4.3)
Venous thrombosis	123 (2.7)
Arterial thrombosis	74 (1.6)
Revision due to wound complications, *n* (%)	378 (8.3)
Infection donor site	23 (0.5)
Infection recipient site	20 (0.4)
Hematoma donor site	37 (0.8)
Hematoma recipient site	148 (3.2)
Wound-healing disturbances at donor site	80 (1.7)
Wound-healing disturbances at recipient site	70 (1.5)
Medical complications, *n* (%)	294 (6.4)

*n*, number.

**Table 5 jcm-10-01016-t005:** Perioperative details according to the average number of DIEP flaps performed per year per center (high-volume group: ≥Ø 40 DIEP flaps per year of data entry vs. low-volume group < Ø 40 DIEP flaps per year of data entry).

Variable	LV Centers	HV Centers	*p* Value
Patients, *n*	1260	2653	
Free flaps, *n*	1459	3118	
Operation time (min)			
Mean (SD)	385.82 (142.31)	287.14 (107.01)	<0.001
Ischemia time (min)			
Mean (SD)	51.81 (27.36)	50.34 (25.12)	0.073
Recipient, *n* (%)			<0.001
Internal mammary	1312 (89.9)	2371 (76.0)	
Thoracodorsal	60 (4.1)	644 (20.7)	
Other	87 (6.0)	103 (3.3)	
Postoperative mobilization, *n* (%)			<0.001
Day 1	472 (32.4)	2821 (90.6)	
Day 2	488 (33.5)	285 (9.1)	
Day 3	118 (8.1)	8 (0.3)	
Day 4	166 (11.4)	0 (0.0)	
Day 5	98 (6.7)	0 (0.0)	
Day 6	72 (4.9)	0 (0.0)	
Day 7	42 (2.9)	0 (0.0)	
LOS, days			
Mean (SD)	9.04 (18.87)	8.21 (5.04)	0.023

Percentages calculated based on the number of free flaps. *n*, number; SD, standard deviation; min, minutes; LOS, length of hospital stay; LV, low-volume; HV, high-volume.

**Table 6 jcm-10-01016-t006:** Postoperative complications according to the average number of DIEP flaps performed per year per center (high-volume group: ≥Ø 40 DIEP flaps per year of data entry vs. low-volume group < Ø 40 DIEP flaps per year of data entry).

Variable	LV Centers	HV Centers	*p* Value
Free flaps, *n*	1459	3118	
Total flap loss, *n* (%)	18 (1.2)	74 (2.4)	0.014
Partial flap loss, *n* (%)	20 (1.4)	31 (1.0)	0.327
Emergent vascular revision surgery, *n* (%)	58 (4.0)	139 (4.5)	0.453
Venous thrombosis	38 (2.6)	85 (2.7)	0.889
Arterial thrombosis	20 (1.4)	54 (1.7)	0.437
Revision due to wound complications, *n* (%)	158 (10.8)	220 (7.1)	<0.001
Infection donor site	9 (0.6)	14 (0.4)	0.6
Infection recipient site	6 (0.4)	14 (0.4)	1
Hematoma donor site	15 (1.0)	22 (0.7)	0.338
Hematoma recipient site	56 (3.8)	92 (3.0)	0.136
Wound-healing disturbances donor site	33 (2.3)	47 (1.5)	0.09
Wound-healing disturbances recipient site	39 (2.7)	31 (1.0)	<0.001
Medical complications, *n* (%)	105 (7.2)	189 (6.1)	0.144

Percentages calculated based on the number of free flaps. *n*, number; LV, low-volume; HV, high-volume.

## Data Availability

3rd Party Data; Restrictions apply to the availability of these data. Data was obtained from the DGPRÄC national online registry for microsurgical breast reconstructions.

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
