# Peer review of "Overall Complication Rates of DIEP Flap Breast Reconstructions in Germany—A Multi-Center Analysis Based on the DGPRÄC Prospective National Online Registry for Microsurgical Breast Reconstructions"

_jcm, 2021, doi:10.3390/jcm10051016_

Round 1
Reviewer 1 Report
It was interesting for me to know about DGPRÄC. I think this paper could be a starting point to reach a common high standard level in autologous breast reconstruction not only in Germany, highlighting everything is needed and everything is to avoid to reach a constant high result.
This obviously is easier for an elective surgery but it could be the base also for other kind of microsurgical job.
Author Response
Response: We thank the reviewer for the extensive review of our manuscript and the positive feedback provided.
Action: None
Reviewer 2 Report
The present study investigated overall outcome of DIEP flap reconstructions in Germany; it was performed using the breast reconstruction prospectively maintained database established by the German Society of Plastic, Reconstructive and Aesthetic Surgeons (DGPRÄC). Data of 3926 patients and 4577 DIEP flaps performed by 22 centers were included in this study.
The number of patients included is relevant; the conclusions of the study highlight the high standard of microsurgical breast reconstruction on a national level in Germany.
The authors hope to increase visibility and transparency of the standard and quality of care in breast reconstruction. I believe that data from the DGPRÄC national online registry for microsurgical breast reconstruction and the rigorous method of German colleagues should be spread to the other health systems; anyway, in my opinion, the manuscript is not of sufficient novelty.
Author Response
Response: We thank the reviewer for the important feedback. The authors do believe in the novelty of this study and want to highlight that the manuscript evaluates plastic surgical breast reconstruction in a national setting in Germany,i.e., it reaches beyond individual specialized clinics, and places it into perspective of current health policy in our field of expertise.
Action: None
Reviewer 3 Report
sing the national registry of microsurgical breast reconstruction, the authors analyze the complications and compare them between two groups of high and low case volume of the institution.
Overall complications were low and the high-volume group showed superiority in some outcome measures.
Studies using the same database have been published as referenced #18-21, which can affect the value of this study for a separate publication.
Bulge and hernia are the most critical complications of the diep flap. The authors should explain why they were excluded from the study by limiting the postoperative period and acknowledge this point.
Author Response
Response: As mentioned correctly by the reviewer, the DGPRÄC database yielded an extensive amount of data, which cannot be sufficiently displayed in a single manuscript. All previously published manuscripts have been cited within this study, and investigated unique and specific outcome parameters. The presented study differs significantly from the previously published manuscripts and presents a novel analysis of the data.
We agree with the reviewer that bulge and hernia are significant donor site complications of DIEP flap reconstruction and evaluation would have added further impact to the study. Hernia and bulging were initially included as an item in the online registry, however the evaluation was problematic, as the definition is vague and could not be effectively verified by the audit team. The design of the registry, including the three months follow-up period were chosen due to legal and financial issues which did not support extensive follow-up patients. In addition, only an inconclusive number of patients were willing to present for a final examination after 12 months.
Action: The discussion section has been revised. Please find changes on page 17/22, lines 279-282.
Round 2
Reviewer 2 Report
Dear Authors, I read your manuscript with great interest and I have to congratulate for the high standard of microsurgical breast reconstruction on a national level in Germany. Such an outstanding result is made both by the surgical skills and by the hospitals' organization.
Anyway I do not see any "conclusions" section in your study and I am not able to find any scientific conclusions of your study
Author Response
Response: We thank the reviewer for acknowledging our work and highlighting the missing conclusions.
Action: Please find conclusions on page 18/22, lines 290-294.
Reviewer 3 Report
The point raised by the reviewer was addressed by the authors.
Author Response
Thank you!